# An Efficient AdaBoost Algorithm with the Multiple Thresholds Classification

Yi Ding [1], Hongyang Zhu [2,*], Ruyun Chen [2] and Ronghui Li [1]

1  Maritime College, Guangdong Ocean University, Zhanjiang 524091, China; dingyi@gdou.edu.cn (Y.D.); lirh@gdou.edu.cn (R.L.)
2  College of Mathematics and Computer, Guangdong Ocean University, Zhanjiang 524091, China; chenry@gdou.edu.cn
*  Correspondence: zhuhongyang@gdou.edu.cn

**Featured Application: A new Weak Learn algorithm which classifies examples is proposed based on multiple thresholds. The weight assigning scheme of the Weak Learn algorithm is changed correspondingly for the AdaBoost algorithm in this paper. Theoretical identification is provided to show the superiority. Experimental studies are also presented to verify the effectiveness of the method.**

**Abstract:** Adaptive boost (AdaBoost) is a prominent example of an ensemble learning algorithm that combines weak classifiers into strong classifiers through weighted majority voting rules. AdaBoost's weak classifier, with threshold classification, tries to find the best threshold in one of the data dimensions, dividing the data into two categories-1 and 1. However, in some cases, this Weak Learning algorithm is not accurate enough, showing poor generalization performance and a tendency to over-fit. To solve these challenges, we first propose a new Weak Learning algorithm that classifies examples based on multiple thresholds, rather than only one, to improve its accuracy. Second, in this paper, we make changes to the weight allocation scheme of the Weak Learning algorithm based on the AdaBoost algorithm to use potential values of other dimensions in the classification process, while the theoretical identification is provided to show its generality. Finally, comparative experiments between the two algorithms on 18 datasets on UCI show that our improved AdaBoost algorithm has a better generalization effect in the test set during the training iteration.

**Keywords:** AdaBoost; Multiple Thresholds Classification; accuracy; generalization

## 1. Introduction

The rapid growth of the Internet has led to a dramatic increase in the rate of data generation. Data mining technology is one of the most important means of mining value from such a large amount of data. Classification is the initialization operation that processes the digitized information of data mining. Obviously, accurate classification will save a lot of time and economic costs for subsequent work, such as analysis, forecasting, and fitting processes.

Ensemble methods are ideal for regression and classification, and by combining multiple models into a very reliable model, it can reduce bias and variance to boost the accuracy of predictions [1,2]. Two common techniques for constructing Ensemble classifiers are Boosting [3–7] and Bagging [8–10]. Boosting is better than Bagging, with less noise in the data [11].

In the field of machine learning, the boosting algorithm is a more classic general-purpose learning algorithm, which is based on the "probably approximately correct" learning model proposed by Valiant. Freund and Schapire improved the Boosting algorithm in 1995 and named it the AdaBoost algorithm.

The AdaBoost algorithm has now developed into an important feature classification algorithm in machine learning, which has been widely used in face detection [12] and image retrieval [13], intrusion detection [14], object recognition [15], feature extraction [16], and other applications.

In the training phase, the weights assigned to the samples by AdaBoost rise as the error rate increases, and conversely, the newly assigned weights fall as the error rate decreases. The samples are then continuously trained with unknown distribution weights. The aim is to obtain powerful feedback by reducing errors on the next machine and ultimately achieving better accuracy [17].

How to design an effective classifier [18] to improve the accuracy of classification is at the core of this type of algorithm. In previous years, the AdaBoost algorithm has attracted great interest of scientists, aiming at the problems of insufficient correct classification rate and long training time, a variety of improved algorithms were proposed [19,20].

One of the scholars focused on solving the problem of unbalanced data classification [21,22] to improve the classification accuracy of the algorithm. Juan J. Rodrigue and Jesus Maudes proposed a new reconfigurable Weak Learn algorithm based on the decision tree algorithm [23]. Utkin L V and Zhuk Y A analyzed a robust classification algorithm called Robust Imprecise Local Boost (RILBoost), which focused on the precise vector of weights assigned for examples in the training set at each iteration of boosting improved classification performance [24]. Another class of scholars has designed a variety of important and influential AdaBoost variants under the framework of forward Additive Stagewise models based on nonparametric regression, including LogitBoost [25], GradientBoost [26], Gradient-boost-tree [26], and ProbitBoost [27] for solving probabilistic regression. In these experiments, they demonstrated that the AdaBoost-based approach provided better detection performance than other non-AdaBoost-based methods.

However, algorithms are difficult to coexist with high classification accuracy and low complexity, that is, strong classifiers are accurate but difficult to obtain, while weak classifiers are the opposite. The Boosting algorithm requires the Weak Learn algorithm to have an accuracy better than 0.5, which is too strong to be fulfilled, and the number of training examples can have a significant impact on accuracy. Studies have shown that even under ideal circumstances, increasing the number of iterations, AdaBoost will eventually overfit [28], and the sub-classifiers generated at the end of the iteration will have very small effect on improving the generalization performance of the classifier, and there is not only a risk of overfitting, but also a waste of computing power.

Furthermore, an analysis of the diversity of sub-classifiers generated by the AdaBoost algorithm found that in the previous iterations [29,30], each new sub-classifier focused on samples that were difficult for the previous sub-classifier to classify correctly, so there was a high diversity. However, after several iterations, samples that are difficult to classify correctly are likely to always be misclassified, resulting in a sharp decline in diversity as the generated sub-classifiers pay more and more attention to the same batch of samples that cannot be classified correctly.

Although many scholars have made significant contributions in improving the accuracy and generalization rates in previous work, the work has mostly focused on modifications to the algorithmic framework, while few scholars have focused on the tuning of the Weak Learn algorithm itself.

In this paper, we solved the problem of low accuracy and weak generalization of the AdaBoost algorithm when using the threshold classification method as a weak classifier. Our learning model faces two challenges. First, how to mine more exploitation values from all features of training samples to improve the accuracy of threshold classification; second, how to assign weights to different weak classifiers during iteration to build integrated classifiers with strong generalization ability. To solve the above two challenges, we propose a novel AdaBoost algorithm with Multiple Threshold Classification as the weak learn algorithm, which can improve the classification accuracy and generalization ability.

In all, the innovation of this paper is that the threshold classification method is improved by taking the weighted voting results of the three thresholds with the highest accuracy as the prediction results, called Multiple Threshold Classification method, and the method is used as a Weak Learning algorithm for the AdaBoost algorithm. Meanwhile, in order to make the Multiple Threshold Classification method more adaptable to the AdaBoost algorithm, when assigning weights to the weak classifiers, we choose the maximum error rate as the weight parameter in order to obtain more weak classifiers with smaller weights to fully exploit the value of all features of the sample. Finally, we can obtain a better predictive classifier with higher accuracy, generalization ability, and convergence in different types of classification problems in different domains.

The rest of this paper is organized as follows. In Section 2, in order to facilitate the description, we introduce the AdaBoost algorithm and Threshold Classification algorithm as the Weak Learn algorithm of AdaBoost. In Section 3, we illustrate our AdaBoost algorithm with Multiple Thresholds Classification and a new weight distribution in detail. Our AdaBoost algorithm is tested for accuracy and generalization in Section 4 using 12 datasets from UCI. Conclusion and future work are given in Section 5.

## 2. Background

In this section, to facilitate the illustration of our AdaBoost algorithm with Multiple Thresholds Classification, we first introduce AdaBoost algorithm in Section 2.1 and the Threshold Classification algorithm in Section 2.2. AdaBoost with Multiple Thresholds Classification is an ensemble learn classification with the AdaBoost as the frame algorithm, and the Threshold Classification as the Weak Learn algorithm. In this section, we first introduce the AdaBoost algorithm in Section 2.1, in order to easily illustrate our AdaBoost algorithm with Multiple Thresholds Classification; in Section 2.2, we introduce the threshold classification algorithm. AdaBoost with multiple threshold classification is an integrated learning classification, where AdaBoost is used as a frame algorithm and threshold classification is used as a Weak Learning algorithm.

### 2.1. AdaBoost

AdaBoost is an effective ensemble learning algorithm that can take full advantage of a limited number of training examples by updating the weight of training examples. The main steps of AdaBoost are presented in Algorithm 1.

---
**Algorithm 1** AdaBoost

---
**1. Input:** Training dataset: $S = \{(X_1, Y_1), (X_2, Y_2), \cdots, (X_m, Y_m)\}$; Weak Learn algorithm; Ensemble size T.
**2. Initialization:** Initialize the training set with uniform weight distribution
$\omega_i^1 : \omega_i^1 = \frac{1}{m}, i = 1, 2, \cdots, m$.
**3. Do for** $t = 1, 2, \cdots, T$
    (3.1) ***generates*** a weak classifier based on the weak classifier learning algorithm with current weight distribution $\omega^t$;
    (3.2) ***calculate*** the weighted training error $\varepsilon_t$: $\varepsilon_t = \sum_{i=1}^{m} \omega_i^t$, $Y_i \neq h_t(X_i)$;
    (3.3) ***assign*** $h_t$ a weight of $\alpha_t$: $\alpha_t = \frac{1}{2} \ln(\frac{1-\varepsilon_t}{\varepsilon_t})$;
    (3.4) ***update*** the weights of the training examples:
      $\omega_i^{t+1} = \frac{\omega_i^t \cdot \exp(-\alpha_t \cdot y_i \cdot h_t(X_i))}{Z_t}$, where $Z_t = \sum_{i=1}^{m} \omega_i^t \cdot \exp(-\alpha_t \cdot y_i \cdot h_t(X_i))$ is the normalization.
**4. Output:** the ensemble classifier: $f(X) = sign(\sum_{t=1}^{T} \alpha_t \cdot h_t(X))$.

---

In training, dataset $S$, $(X_i, Y_i)$ is the $i$th example with $X_i \in R^d$ showing the attributes and $Y_i \in \{-1, 1\}$ doing the label ($i = 1, 2, \cdots, m$), and the weight distribution over all samples is initially set uniform in step 2. Then, AdaBoost calls Weak Learn algorithm repeatedly in a series of iteration as shown in step 3. In the $t$ th iteration, the Weak Learn trains a classifier $h_t$ and the distribution $\omega^t$ is updated after each iteration according to the prediction results on the training samples. "Easy" samples which are correctly classified by

$h_t$ obtain lower weights, and "hard" samples that are misclassified become higher weights according to step 3.4.

In the next iteration, there will be a new classifier $h_t$ in case of a new distribution $\omega^{t+1}$. Due to this, AdaBoost focuses on the samples with higher weights, which seem to be harder to use for the Weak Learn algorithm. It continues for T iterations, and, finally, AdaBoost linearly combines each component classifier to form a single final hypothesis $f$.

### 2.2. Threshold Classification

By increasing weights of error-classifying examples, AdaBoost focuses the 'hard' (misclassified) examples and trains learners in an iterative way. The final decision is combined by a set of diverse classifiers using weighted majority voting rule. Without the difficulty in directly designing an excellent algorithm, AdaBoost takes full advantage of Weak Learn algorithm, which is easily available. For a great many kinds of training example sets, there are obvious relations between labels and certain attributes, in other words, most examples with larger values of certain attribute fall in the same category, and the ones with smaller values fall in the other category. Based on the relation, University of Twente has written out the program of the classical AdaBoost algorithm, AdaBoost with Threshold Classification, in 2010. Finding the best threshold from all attributes as classification rule, this AdaBoost provides a rough but easily obtainable learner in each iteration. The Threshold Classification algorithm is presented in Algorithm 2.

---

**Algorithm 2** Threshold Classification.

---

1. Input: Training dataset: $S = \{(X_1, Y_1), (X_2, Y_2), \cdots, (X_m, Y_m)\}$; weight distribution $\omega$.
2. Do for $j = 1, 2, \cdots, d$
   (2.1) Sort the $m$ training examples by the size of their $j$th attribute;
   (2.2) Find the threshold $\tau_j$ in the attribute as the classifier $h_j$.
3. Output: Compare these weighted errors, $\varepsilon_1, \varepsilon_2, \cdots, \varepsilon_d$, and find the minimum $\varepsilon = \min\{\varepsilon_j\}(j = 1, 2, \cdots, d)$, the threshold classifier is $h = h_j$ with the error $\varepsilon_j = \varepsilon$.

---

In the train dataset $S$, $(X_i, Y_i)$ is the $i$th example, and $X_i = (x_{i1}, x_{i2}, \cdots, x_{iJ})$ describes the $i$th example $(j = 1, 2, \cdots, d)$, while $Y_i \in \{-1, 1\}$ does the label $(i = 1, 2, \cdots, m)$. So $x_{ij}$ is the $j$th attribute Absolutely of $i$th example and every example is described by $d$ attributes and a label. We find the threshold $\tau_j$ as a classifier $h_j$ from $\{x_{1j}, x_{2j}, \cdots, x_{mj}\}$ which are the $j$th attributes of all the examples so that we obtain the lowest weighted error $\varepsilon_j = \sum_{i=1}^m \omega_i, Y_i \neq h_t(X_i)$ when we distinguish the examples whose $j$th attribute are larger than $\tau_j$ from those that are smaller than $\tau_j$. When the $j$th attributes of most positive examples are larger than $\tau_j$ and that of most negative examples are smaller than $\tau_j$:

$$h_j(X_i) = \begin{cases} 1, & x_{ij} \geq \tau_j \\ -1, & x_{ij} < \tau_j \end{cases} \tag{1}$$

when the $j$th attributes of most positive examples are smaller than $\tau_j$ and that of most negative examples are larger than $\tau_j$: $h_j(X_i) = \begin{cases} 1, & x_{ij} \leq \tau_j \\ -1, & x_{ij} > \tau_j \end{cases}$ in step 2. The accuracy of the weak classifier $h$ based on Threshold Classification is $1 - \varepsilon$.

## 3. AdaBoost with Multiple Thresholds Classification

This section explains the basic idea of our Multiple Thresholds Classification in Section 3.1. Afterwards, we will construct the Weak Learn algorithm in the Algorithm frame and then briefly describe our algorithm in Section 3.2.

### 3.1. Multiple Thresholds Classification

Instead of building the complicated relation between labels and all the attributes, AdaBoost algorithm based on threshold focuses on the liner relation between labels and

the most critical attribute. However, this classifier may achieve a low accuracy and poor generalization due to the following reasons: (1) this Weak Learn algorithm just searches only the one optimal threshold from all the attributes as the classified rule that wastes the potential value of other resources; (2) this classifier still has the tendency of overfitting as it focuses on training examples which is misclassified by just the one threshold in each iteration. If the labels tend to have certain degree linear relations with not only one attribute, such as the relation between gender and height or weight, AdaBoost algorithm with Threshold Classification may achieve not only a low accuracy but also poor generalization. For example, if we classify persons by height, like that the taller persons are divided into males and shorter ones into females, there must be many short males and tall females misclassified, the same as by weight. To solve the dilemma, we can divide those who are tall and heavy into males and the others into females, so there will be less persons misclassified. To improve the Threshold Classification, we proposed the Multiple Thresholds Classification which uses three crucial thresholds as a classifier, not only the one. Firstly, we sort examples in sequences by attributions, and choose thresholds from every sequence, the same as step 2 of Algorithm 2. According to the errors of these thresholds, we choose the best three ones and assign different weights to them, then we integrate those three classifiers by weighted majority voting rule as a Weak Learn algorithm. It is formally demonstrated in Algorithm 3.

---

**Algorithm 3** Multiple Thresholds Classification.

---

**1. Input:** Training dataset: $S = \{(X_1, Y_1), (X_2, Y_2), \cdots, (X_m, Y_m)\}$, weight distribution $\omega$.

**2. Do for** $j = 1, 2, \cdots, d$

(2.1) Sort the $m$ training examples by the size of their $j$th attribute;

(2.2) Find the threshold $\tau_j$ in the $j$th attribute as the classifier $h_j$ so that we obtain the lowest weighted error $\varepsilon_j = \sum_{i=1}^{m} \omega_i, Y_i \neq h_t(X_i)$ as the step (2.2) in Algorithm 2.

**3. Output:** Compare weighted errors $\varepsilon_1, \varepsilon_2, \cdots, \varepsilon_d$ of these classifiers, $h_1, h_2, \cdots, h_d$, to find the three best classifiers $h', h'', h'''$ with the lowest errors $\varepsilon', \varepsilon'', \varepsilon'''$, which $0 \leq \varepsilon' \leq \varepsilon'' \leq \varepsilon''' \leq 1$.

The prediction result is weighted majority voted by the three rules:

$h(X) = sign \{(1 - \varepsilon')h'(X) + (1 - \varepsilon'')h''(X) + (1 - \varepsilon''')h'''(X)\}$

Weighted error of $h(X)$ is $e = \sum_{i=1}^{m} \omega_i^t, Y_i \neq h(X_i)$.

---

The error $\varepsilon$ of a classifier means that the classifier will classify an example correctly with the probability of $1 - \varepsilon$, so in Algorithm 3, the weak classifier $h$ with multiple thresholds will classify examples correctly in the following five cases:

1. all the three classifiers $h'$, $h''$, $h'''$ classify correctly, $h'(X) = Y$, $h''(X) = Y$, $h'''(X) = Y$ according to Equation (1), there is $h(X) = Y$, and the probability is:

   This is example 1 of an equation:

$$p_1 = (1 - \varepsilon')(1 - \varepsilon'')(1 - \varepsilon''') \tag{2}$$

2. $h', h''$ classify correctly, but $h'''$ not, $h'(X) = Y$, $h''(X) = Y$, $h'''(X) \neq Y$, $h(X) = Y$ in that $\varepsilon' \leq \varepsilon'' \leq \varepsilon'''$ and the probability is:

$$p_2 = (1 - \varepsilon')(1 - \varepsilon'')\varepsilon''' \tag{3}$$

3. $h''$, $h'''$ classify correctly, but $h''$ not. The probability is:

$$p_3 = (1 - \varepsilon')\varepsilon''(1 - \varepsilon''') \tag{4}$$

4. $h''$, $h'''$ classify correctly, but $h'$ not and the sum of accuracies of $h''$ and $h'''$ is larger than that of $h'$, $(1 - \varepsilon'') + (1 - \varepsilon''') \geq (1 - \varepsilon')$, in other words, $\varepsilon'' + \varepsilon''' - \varepsilon' \leq 1$. The probability is:

$$p_4 = \varepsilon'(1-\varepsilon'')(1-\varepsilon''') \cdot p \tag{5}$$

$p$ is the probability that $\varepsilon'' + \varepsilon''' - \varepsilon' \leq 1$.

5.  $h'$ classifies correctly, but $h'', h'''$ not and the sum of accuracies of $h''$ and $h'''$ is smaller than that of $h'$, $(1-\varepsilon'') + (1-\varepsilon''') \leq (1-\varepsilon')$, in other words, $\varepsilon'' + \varepsilon''' - \varepsilon' \geq 1$. The probability is:

$$p_5 = (1-\varepsilon')\varepsilon''(1-\varepsilon''') \cdot (1-p) \tag{6}$$

According to Equations (2)–(6), expectation of accuracy of Multiple Thresholds Classification $h(X)$ is:

$$\partial = p_1 + p_2 + p_3 + p_4 + p_5 \tag{7}$$

Assume that the errors follow uniform distribution pattern: $\varepsilon' \sim U(0,1)$, $\varepsilon'' \sim U(\varepsilon',1)$, and $\varepsilon''' \sim U(\varepsilon'',1)$, then the joint density function is:

$$f(x,y,z)\big|_{\varepsilon',\varepsilon'',\varepsilon'''} = \begin{cases} \frac{1}{(1-x)(1-y)}, & 1 \geq x \geq y \geq 0 \\ 0, & else \end{cases} \tag{8}$$

$\varepsilon'' + \varepsilon''' - \varepsilon' \geq 1$ and $0 \leq \varepsilon' \leq \varepsilon'' \leq \varepsilon''' \leq 1$ mean that $\max\{1+\varepsilon'-\varepsilon'', \varepsilon''\} \leq \varepsilon''' \leq 1$ while $\varepsilon' \leq \varepsilon'' \leq 1$. $1+\varepsilon'-\varepsilon''$ and $\varepsilon''$ are compared as shown: $\begin{cases} 1+\varepsilon'-\varepsilon'' > \varepsilon'', & as\ \varepsilon'' < \frac{1+\varepsilon'}{2} \\ 1+\varepsilon'-\varepsilon'' \leq \varepsilon'', & as\ \varepsilon'' \geq \frac{1+\varepsilon'}{2} \end{cases}$, besides Equation (8), $\partial$ in Equation (7) is calculated as followed:

$$\partial = \int_0^1 dx \int_x^{\frac{1+x}{2}} dy \int_{1+x-y}^1 \frac{1}{(1-x)(1-y)} dz + \int_0^1 dx \int_{\frac{1+x}{2}}^1 dy \int_y^1 \frac{1}{(1-x)(1-y)} dz = \ln 2$$

The accuracy of threshold classifier $1-\varepsilon$ in Algorithm 2 is equal to the accuracy of Multiple Thresholds Classifier $1-\varepsilon'$ in Algorithm 3. If $\varepsilon' \sim U(0,1)$ expectation of $\varepsilon'$ is 50%, so is $1-\varepsilon'$. Obviously, expectation of accuracy of Multiple Thresholds Classification $\partial$ in Algorithm 4 is better than that of Threshold Classification $1-\varepsilon'$.

---

**Algorithm 4** AdaBoost with Multiple Thresholds Classification.

---

**1. Input:** Training dataset $S$ and Ensemble size T as the same as in Algorithm 1; Multiple Thresholds Classification.

**2. Initialization:** the same as in Algorithm 1;

**3. Do for** $t = 1, 2, \cdots, T$

(3.1) *generates* a weak classifier $h_t(X)$ as shown in Algorithm 3.

(3.2) the greater the error $\varepsilon_t$ of learner $h_t(X)$ is, the smaller the weight of $h_t(X)$ will be, so we prefer to assign a severe weight to $h_t(X)$ by choosing a greater error $\varepsilon_t$ between $\varepsilon'''$ and $e$ in Algorithm 3: $\varepsilon_t = max\{\varepsilon''', e\}$. The weight of $h_t(X)$ is: $\alpha_t = \frac{1}{2}\ln(\frac{1-\varepsilon_t}{\varepsilon_t})$.

(3.3) *Reassign* weights for the training examples: $\omega_{t+1}(i) = \frac{\omega_t(i) \cdot e^{-\alpha_t \cdot y_i \cdot h_t(X_i)}}{Z_t}$,

$Z_t = \sum_{i=1}^m \omega_t(i) \cdot e^{-\alpha_t \cdot y_i \cdot h_t(X_i)}$ is the normalization.

**4. Output:** the ensemble classifier: $f(X) = sign\left\{ \sum_{t=1}^T \alpha_t \cdot h_t(X) \right\}$.

---

### 3.2. Multiple Thresholds Classification as the Weak Learn Algorithm

Although a weighted majority voting rule can prevent the final decision from overfitting effectively, the Weak Learn algorithm of AdaBoost with Threshold Classification still has the tendency of overfitting when the adjacent Weak Learn algorithm is from the same attribute. We try to enhance the generalization of the final decision by enhancing that of every Weak Learn algorithm. In AdaBoost with Multiple Thresholds Classification, (1) we choose multiple thresholds as a Weak Learn algorithm to avoid monopoly and exploit more

potential value of the whole training set; (2) in the case where the Threshold Classification is not accurate as Weak Learn algorithm of the AdaBoost, we prefer to make more but 'smaller' Weak Learn algorithms, so we improve a new mode to assign smaller weights to weak classifiers.

The only determinant of $\alpha$ is the error $\varepsilon$ in (3.3) in Algorithm 4, so we can obtain smaller $\alpha$ by choosing a greater $\varepsilon$. There appear four learners including $\varepsilon'$, $\varepsilon''$, $\varepsilon'''$ and $e$, and any one of their weighted errors can be used in (3.3) theoretically. Obviously, the weighted error $\varepsilon'''$ is the greatest in $\varepsilon'$, $\varepsilon''$, $\varepsilon'''$, so we choose the greater one from $\varepsilon'''$ and $e$ as the $\varepsilon$ in Algorithm 4.

## 4. Experiment

We compare our AdaBoost based on Multiple Thresholds Classification with classical AdaBoost by conducting experiments on 18 multi-class classification problems in Section 4, so that the effectiveness of the method can be demonstrated. These adopted datasets used in this study are popular binary classification datasets with numerical attributes and few incomplete instances from various areas in UCI repositories.

### 4.1. Data Set Information and Parameter Setting

We carried out the program of AdaBoost with Multiple Thresholds Classification in MATLAB R2014a and adopted datasets from the UCI machine learning repository listed in Table 1 as experiments. To test generalization ability of the new algorithm, those datasets are diversified. They are chosen from various areas, life, game, physical, business, computer, computer security and so on, with different amounts of instances and attributes. Getting rid of those incomplete examples, the smallest size of the 18 datasets is 90, while the largest is 17,898, and the examples have at least 6 attributes and 101 at most. Detailed information about these datasets can be found in https://archive.ics.uci.edu/ml/index.php (accessed on 26 May 2022). For each dataset, 70 percent of examples are randomly extracted as the training subset and 30 percent as the testing subset if providers do not give advisement. The weight of examples and weak classifiers are automatically assigned according to the classification result, so there is only one parameter that needs to be set for AdaBoost Algorithms, the maximal number of iterations, which is set as 100 that is acceptable for computing power and enough for the most datasets to converge.

**Table 1.** Descriptions of the datasets.

| No. | Dataset | Number of Web Hits: | Number of Examples | Number of Attributes |
|---|---|---|---|---|
| 1 | Abalone | 1,268,791 | 4177 | 8 |
| 2 | Breast Cancer Wisconsin (Diagnostic bcw) | 1,742,253 | 683 | 10 |
| 3 | Breast Cancer Wisconsin (Diagnostic wdbc) | 1,742,253 | 569 | 32 |
| 4 | Connectionist Bench (Sonar) | 237,567 | 208 | 60 |
| 5 | Cryotherapy | 62,013 | 90 | 7 |
| 6 | EEG Eye State | 154,643 | 14,980 | 15 |
| 7 | Hill-Valley | 79,654 | 1212 | 101 |
| 8 | HTRU2 | 87,855 | 17,898 | 9 |
| 9 | Immunotherapy | 69,214 | 90 | 8 |
| 10 | Ionosphere | 286,248 | 351 | 34 |
| 11 | Liver Disorders | 216,408 | 345 | 7 |
| 12 | Molecular Biology (Splice) | 116,402 | 3190 | 61 |
| 13 | Raisin | 1,305,031 | 900 | 8 |
| 14 | seismic-bumps | 82,013 | 2584 | 19 |
| 15 | SPECTF Heart | 111,087 | 267 | 44 |
| 16 | Statlog (Heart) | 277,569 | 270 | 13 |
| 17 | Wholesale Customers | 435,201 | 440 | 8 |
| 18 | Wine Quality | 1,875,937 | 6497 | 12 |

*4.2. Wilcoxon Rank-Sum Test of These Two Algorithms and Analysis of Experimental Results*

Comparisons of these two algorithms on 18 testing subsets are shown in Table 2. T_Ad is the abbreviation of AdaBoost with Threshold Classification, and MT_Ad is the abbreviation of AdaBoost with Multiple Thresholds Classification, in both Figures 1 and 2. From Table 2, AdaBoost based on Multiple Thresholds Classifications perform significantly better for most of the 18 datasets with different numbers of examples and attributes.

**Table 2.** The accuracy of the two algorithms.

| No. | Dataset | T_Ad | MT_Ad |
|---|---|---|---|
| 1 | Abalone | 0.8138 | 0.8013 |
| 2 | Breast Cancer Wisconsin (Diagnostic bcw) | 0.9854 | 0.9756 |
| 3 | Breast Cancer Wisconsin (Diagnostic wdbc) | 0.9532 | 0.9649 |
| 4 | Connectionist Bench (Sonar) | 0.6774 | 0.7097 |
| 5 | Cryotherapy | 0.8148 | 0.8889 |
| 6 | EEG Eye State | 0.5642 | 0.5735 |
| 7 | Hill-Valley | 0.5495 | 0.5611 |
| 8 | HTRU2 | 0.9868 | 0.9883 |
| 9 | Immunotherapy | 0.7778 | 0.7778 |
| 10 | Ionosphere | 0.9524 | 0.9524 |
| 11 | Liver Disorders | 0.7212 | 0.7404 |
| 12 | Molecular Biology (Splice) | 0.9587 | 0.9630 |
| 13 | Raisin | 0.8852 | 0.8704 |
| 14 | Seismic-bumps | 0.9665 | 0.9665 |
| 15 | SPECTF Heart | 0.7059 | 0.7540 |
| 16 | Statlog (Heart) | 0.8395 | 0.8519 |
| 17 | Wholesale Customers | 0.8864 | 0.9167 |
| 18 | Wine Quality | 0.9932 | 0.9875 |

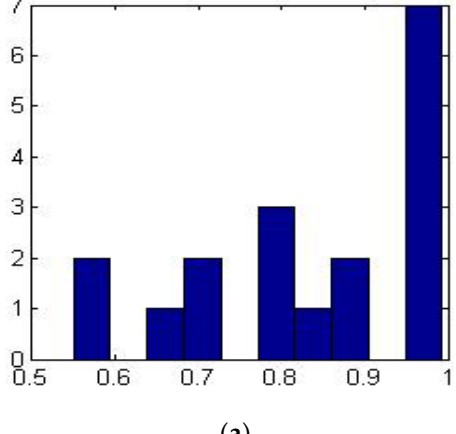 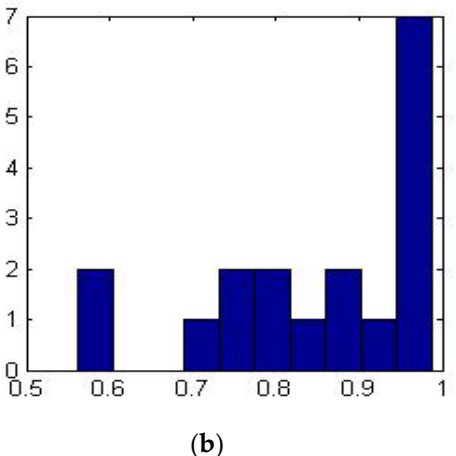

(**a**)          (**b**)

**Figure 1.** Frequency histogram of accuracies of two algorithms: (**a**) AdaBoost with Threshold Classification; (**b**) AdaBoost with Multiple Thresholds Classification.

Figure 1 shows frequency histogram of accuracies of these two algorithms. Obviously, accuracies of these two algorithms are not consistent with a normal distribution, so Wilcoxon signed-rank test is set to prove significant difference between the two algorithms.

Null hypothesis and alternative hypothesis are set as:

$H_0$: there is no significant difference between the two algorithms;

$H_1$: AdaBoost with Multiple Thresholds Classification is better than that with Threshold Classification.

The probability of $H_0$, $p(H_0) \approx 0.0078 \leq 0.05$ by Wilcoxon signed-rank test, so the null hypothesis $H_0$ was rejected at the significance level 0.05, in other words, $H_1$ is received. Specifically, AdaBoost based on Multiple Thresholds Classification performs better than that based on Threshold Classification by the result of Wilcoxon signed-rank test.

Figure 2 shows the variation process of this two algorithms' convergence performances directly. In each panel, we depict the changes of training sets errors and testing sets errors with the growth of iterations of the 18 datasets, abscissa, and ordinate values indicate times of training iteration and errors of two classifiers, respectively. The blue straight line with triangles indicates the test data's error of the test data, and the dashed line with a cross indicates the error of the training data of the threshold classification method AdaBoost; the red line with a triangle indicates the error of the test data, and the dashed line with a cross indicates the error of the training data of the AdaBoost with Multiple Thresholds Classification.

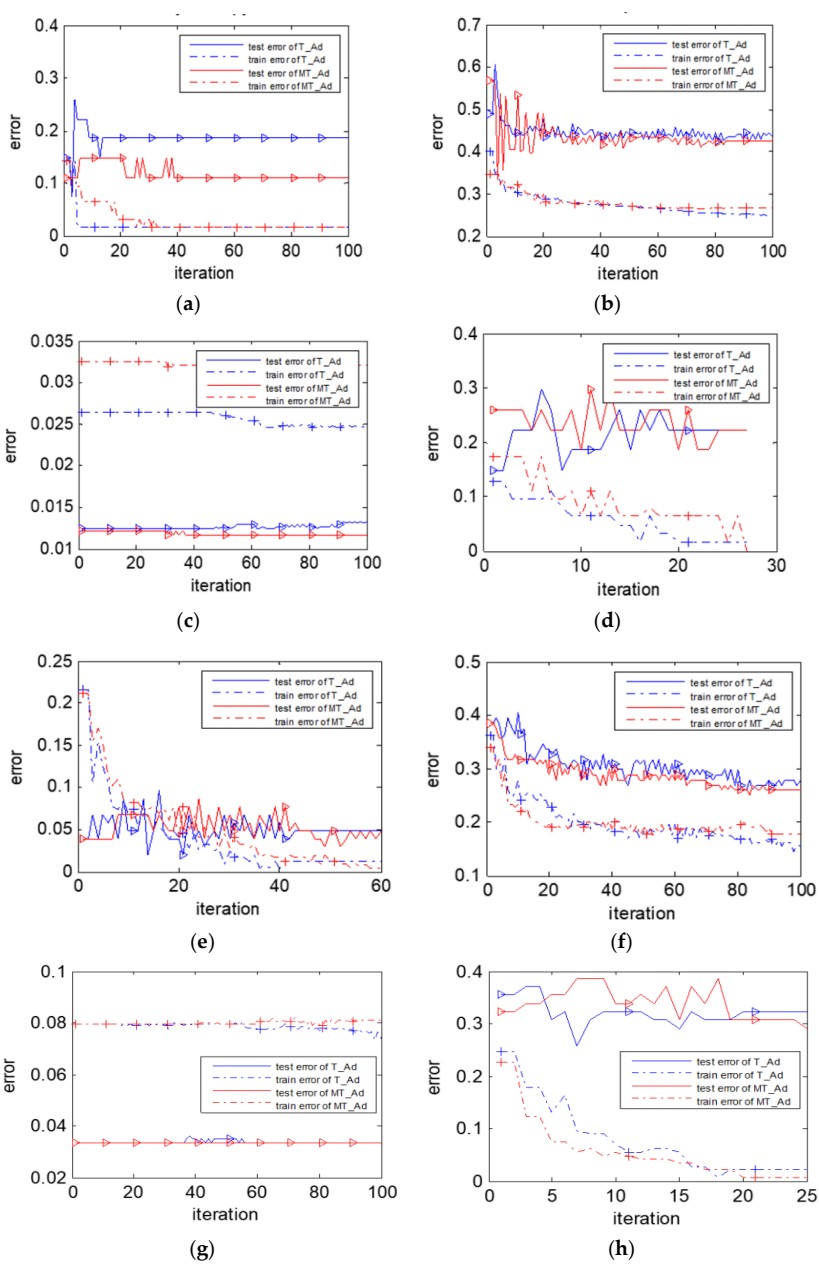

**Figure 2.** *Cont.*

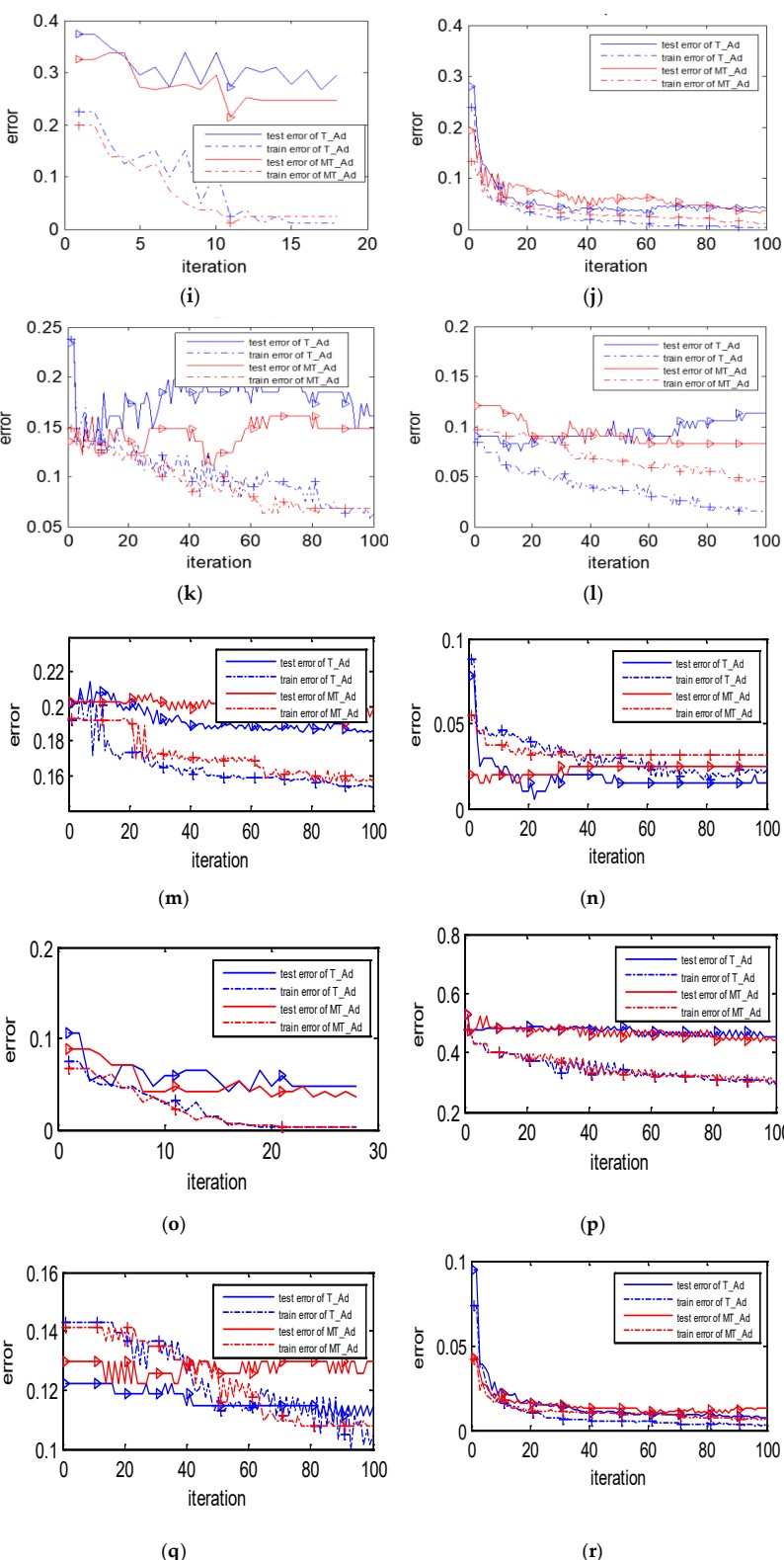

**Figure 2.** Comparison of the two algorithms for datasets (18 datasets). (**a**) Cryotherapy dataset; (**b**) EEG eye state dataset; (**c**) HTRU2 dataset; (**d**) Immunotherapy dataset; (**e**) Ionosphere dataset; (**f**) Liver Disorders dataset; (**g**) Seismic-bumps dataset; (**h**) Sonar dataset; (**i**) Spectf dataset; (**j**) Splice dataset; (**k**) Statlog (Heart) dataset; (**l**) Wholesale Customers dataset; (**m**) Abalone dataset; (**n**) Breast Cancer Wisconsin (Diagnostic bcw) dataset; (**o**) Breast Cancer Wisconsin (Diagnostic wdbc) dataset; (**p**) Hill-Valley dataset; (**q**) Raisin dataset; (**r**) Wine Quality dataset.

### 4.3. Analysis of Experimental Results

By Table 2, it is obvious that the new AdaBoost is better than the classical one for test subsets of Breast Cancer Wisconsin (Diagnostic wdbc) dataset, Connectionist Bench (Sonar) dataset, Cryotherapy dataset, EEG Eye State dataset, Hill-Valley dataset, HTRU2 dataset, Liver Disorders dataset, Molecular Biology (Splice) dataset, SPECTF Heart dataset, Statlog (Heart) dataset, and Wholesale Customers dataset.

From Figure 1, we can learn that the blue dotted liners are lower than the red ones generally; it means that the AdaBoost with Threshold Classification is roughly better than AdaBoost with Multiple Thresholds Classification on accuracy in training sets, but not in testing sets in case the blue full lines are higher than the red ones for datasets of Cryotherapy, EEG eye state, HTRU2, Immunotherapy, Ionosphere, Liver Disorders, Seismic-bumps, Sonar, Spectf, Wholesale Customers, Breast Cancer Wisconsin (Diagnostic wdbc), and Hill-Valley. It indicates that the improved algorithm is with better generalization performance. To discuss the setting where the proposed algorithm performs better, datasets are sorted in ascending order by accuracies of AdaBoost with Threshold Classification, and these datasets are labeled as "1" with the accuracies of AdaBoost with Multiple Thresholds Classification being higher than that of AdaBoost with Threshold Classification, and the others are labeled as "−1" as shown in Table 3.

**Table 3.** The rankings and labels.

| No. | Dataset | Ranking | Label |
|-----|---------|---------|-------|
| 1 | Hill-Valley | 1 | 1 |
| 2 | EEG Eye State | 2 | 1 |
| 3 | Connectionist Bench (Sonar) | 3 | 1 |
| 4 | SPECTF Heart | 5 | 1 |
| 5 | Liver Disorders | 4 | 1 |
| 6 | Immunotherapy | 6 | 1 |
| 7 | Abalone | 7 | −1 |
| 8 | Cryotherapy | 8 | 1 |
| 9 | Statlog (Heart) | 9 | 1 |
| 10 | Raisin | 10 | −1 |
| 11 | Wholesale Customers | 11 | 1 |
| 12 | Ionosphere | 12 | 1 |
| 13 | Breast Cancer Wisconsin (Diagnostic wdbc) | 13 | 1 |
| 14 | Molecular Biology (Splice) | 14 | 1 |
| 15 | Seismic-bumps | 15 | −1 |
| 16 | Breast Cancer Wisconsin (Diagnostic bcw) | 16 | −1 |
| 17 | HTRU2 | 17 | 1 |
| 18 | Wine Quality | 18 | −1 |

Correlation coefficient between rankings and labels is $-0.4423 < -0.3$, which indicates a weak but nevertheless effective linear relationship. So, without considering other factors, such as numerical values of attributes, areas of datasets and so on, AdaBoost with Multiple Thresholds Classification seems to be more suitable on those "hard" datasets whose accuracies is low by classical AdaBoost. It is because the "hard" datasets obviously contain more "hard" examples, and Multiple Thresholds Classification can classify those "hard" examples more accurately than Threshold Classification.

## 5. Conclusions

In this paper, in order to improve the accuracy and generativity of AdaBoost with Thresholds Classification method and avoid the generation of overfitting and voting monopoly, we proposed the AdaBoost Multiple Thresholds Classification algorithm and selected three crucial attributes in each iteration to build a new Weak Learn algorithm. Accurate decisions were obtained using changes in the weighting scheme during the iter-

ations, which positively affected the final ensemble decision of AdaBoost with Multiple Thresholds Classification.

In this study, we directly compared the AdaBoost algorithm with the AdaBoost multi-threshold classification method by working on 18 datasets from UCI. The experimental data results clearly show that better performance is achieved in our AdaBoost based on the modified AdaBoost macro framework while modifying the specific Weak Learning algorithm. It was a little bit tedious to set the parameters of each algorithm. In future, we are still trying to improve the efficiency of AdaBoost Multiple Thresholds Classification and find the best way to use our algorithm for multi-class scenarios.

**Author Contributions:** Conceptualization, Y.D. and H.Z.; methodology, Y.D.; software, H.Z.; validation, H.Z., R.C. and R.L.; data curation, H.Z.; writing—original draft preparation, Y.D.; writing—review and editing, H.Z.; supervision, R.C. All authors have read and agreed to the published version of the manuscript.

**Funding:** This research was funded by the Program for Scientific Research Start-up Funds of Guangdong Ocean University, grant number: R17015.

**Institutional Review Board Statement:** Not applicable.

**Informed Consent Statement:** Not applicable.

**Data Availability Statement:** Not applicable.

**Conflicts of Interest:** The authors declare that they have no known competing financial interest or personal relationship that could have appeared to influence the work reported in this paper.

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
