# Peer review of "An Efficient AdaBoost Algorithm with the Multiple Thresholds Classification"

_applsci, doi:10.3390/app12125872_

Round 1

Reviewer 1 Report

The authors can further improve the significance of the paper by:

1. Explaining why did they choose the particular set of datasets for the experiments.

2. Perform a statistical test (e.g., Wilcoxon or T test) to compare the results of the proposed implementation of AdaBoost and the classical one.

3. Add a discussion section where the results are synthesized and discussed - what are the improvements per dataset in terms of accuracy, or speed.

Author Response

Dear Reviewer,

Thank you very much for your time involved in reviewing the manuscript and your very encouraging comments on the merits.

Please see the attachment for details of the modifications.

To facilitate this discussion, we first retype your comments in italic font and then present our responses to the comments in this cover letter.

Sincerely, daisy ding

Reviewer 2 Report

The paper proposes an AdaBoost Algorithm based on multiple thresholds classification. The paper is well-written and the results are encouraging. Some points for improvement are listed below:

  • The results are based on 12 datasets. In my opinion, the small number of datasets is rather small for producing safe results for the efficiency of the method. 
  • A statistical test should be provided for comparing the produced results between the methods.
  • What was the parameter setting of the algorithms used in the study?
  • A discussion section should be included analyzing under which settings the proposed algorithm is predominant.

Author Response

(The authors gave the same response as above.)

Round 2

Reviewer 1 Report

The authors have addressed my comments.